# AND: Adversarial Neural Degradation for Learning Blind Image Super-Resolution

**Fangzhou Luo**
McMaster University
luof1@mcmaster.ca

**Xiaolin Wu**\*
McMaster University
xwu@ece.mcmaster.ca

**Yanhui Guo**
McMaster University
guoy143@mcmaster.ca

## Abstract

Learnt deep neural networks for image super-resolution fail easily if the assumed degradation model in training mismatches that of the real degradation source at the inference stage. Instead of attempting to exhaust all degradation variants in simulation, which is unwieldy and impractical, we propose a novel adversarial neural degradation (AND) model that can, when trained in conjunction with a deep restoration neural network under a minmax criterion, generate a wide range of highly nonlinear complex degradation effects without any explicit supervision. The AND model has a unique advantage over the current state of the art in that it can generalize much better to unseen degradation variants and hence deliver significantly improved restoration performance on real-world images.

## 1 Introduction

Deep learning has made great strides in the applications of image restoration. It has demonstrated superior performances over traditional methods on almost all common image restoration tasks, including super-resolution [8], denoising [52], compression artifacts removal [6], deblurring [38], etc. But the margin of performance gains made by deep learning methods of image restoration decreases sharply if the degradation processes assumed in training mismatch those of the real world images at inference stage [3]. It is well known that, for any real-world problems, the efficacy of a machine learning technique relies not only on the design of the technique itself but also, sometimes even more critically, on the statistical agreement between the training and test data [42].

In reality, it is either intractable or highly expensive to obtain both degraded images and the corresponding latent images (ground truth). The most common practice in literature is to use a degradation model to generate paired degraded and ground truth images for training the restoration networks [51, 43, 26, 24]. This synthesis approach cannot accurately simulate the realistic digital imaging pipeline that is affected by multiple complex and compounded degradation sources; for instances, insufficient sampling rate, color demosaicing errors, sensor noises, camera jitters, compression distortions and etc. In this paper, we focus on the task of super-resolution, namely assuming that the dominant degradation cause is insufficient sampling rate, which is compounded by other degradation sources in the imaging pipeline. The said complex nonlinear phenomena often defy explicit analytical modeling. A brute force approach may be to build multiple simpler parametric degradation models, one for each type of degradation (e.g., downsampling, noises, compression, motion, etc.) and apply them in different combinations, orders and parameter setting to generate training data, in hope to simulate as wide a range of degradations encountered in practice as possible. This amounts, however, to fighting a losing battle because it is impossible to exhaust all degradation variants, many of which are not even known or understood.

---

\*Corresponding author

37th Conference on Neural Information Processing Systems (NeurIPS 2023).

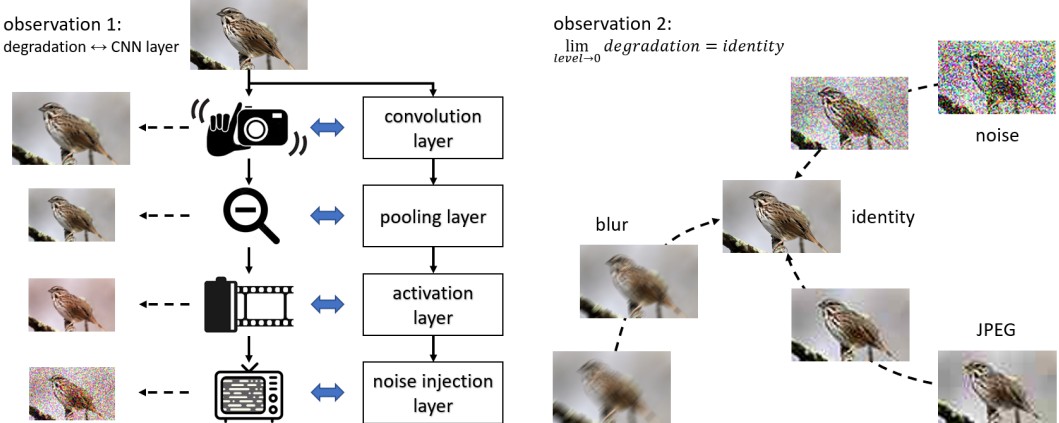

Figure 1: We observe two properties in most image degradations. Firstly, almost all types of image degradation could find a corresponding operation in a standard convolutional neural network. Secondly, almost all moderate image degradations could be considered as small deviations from the identity transformation. The neural degradation prior proposed for our real-world super-resolution method is inspired by these observations.

This work represents a fundamental departure from the current ways of coping with mismatches between the simulated training and real-world image data. Instead of attempting to exhaust all degradation types in simulation, we propose a novel adversarial neural degradation (AND) model that can, when trained in conjunction with a deep restoration neural network under a minmax criterion, generate a wide range of highly nonlinear complex degradation effects without any explicit supervision. Adversarial attack and defense (training) [12, 32] is a proven learning strategy to vaccinate neural network models of signal classification against being misled by imperceptible disturbances in input signals. Previous research in adversarial learning for image restoration tasks [49] only borrows the earlier research in adversarial training for image classification tasks, but does not account for the differences between the classification and restoration tasks. The AND model, the main contribution of this paper, demonstrates for the first time how adversarial learning can effectively boost the robustness of deep networks for signal restoration. In particular, we adopt the minmax optimization criterion when training the AND model, aiming to withstand the attacks by the most difficult but nuance degradations that otherwise defy modeling. As a result, the AND model enjoys a unique advantage over the current state of the art in being generic in terms of degradation types. It can generalize much better to unseen degradation types and variants and hence deliver significantly improved restoration performance on real-world images.

Our insight of the AND model comes from the following observations. We observe two properties in most image degradations, as shown in Fig. 1. Firstly, almost all types of image degradation could find a corresponding operation in a standard convolutional neural network. For example, blur and ringing could correspond to a convolution layer, downsampling could correspond to a pooling layer, color fading and posterization [35] could correspond to a non-linear activation layer, sensor noise and film grain could correspond to a noise injection layer [19]. Secondly, most moderate image degradations could be considered as small deviations from the identity transformation. For example, blur, noise and lossy image compression all obviously tend to the identity transformation pointwise as the degradation level approaches the slightest level. As the degradation level get higher, those degradations gradually deviate from the identity transformation. Inspired by the two observations of image degradations, we initialize untrained convolutional neural networks to the identity transformation, make parameters of these networks slightly deviated from the start, and take them as prior for various real-world image degradations. When we train a SR model with HR and LR image pairs constructed by the proposed degradation prior, we adversarially search small deviations which could make the SR model perform the worst, and optimize the SR model based on the worst degradation case to achieve a good lower performance bound for various real-world image degradations.

## 2 Related Work

**Single Image Super-Resolution.** The first convolutional neural network for single image super-resolution is proposed by Dong et al. [7] called SRCNN, and it achieved superior performance against previous works. Since that the field has witnessed a variety of developments. Shi et al. [36] firstly proposed a real-time super-resolution algorithm ESPCN by proposing the sub-pixel convolution layer. Lim et al. [27] removed batch normalization layers in the residual blocks, and greatly improved the SR effect. Zhang et al. [56] introduced the residual channel attention to the SR framework. To achieve photo-realistic results with detailed textures, Ledig et al. [22] introduced the generative adversarial network [11] into the SR framework, and employed it as loss supervisions to push the SR solutions closer to the natural manifold. Wang et al. [44] later improved the GAN based SR method, and achieved better SR visual quality with more realistic and natural textures.

**Blind Image Super-Resolution.** The field is also named as real-world image super-resolution. Different from the classical SR field which assumes that the image degradation model is an ideal bicubic downsampling, the blind SR field aims to solve SR problems with unknown degradation. Researchers tried to solve the problem by implicitly or explicitly estimating the degradation model. Gu et al. [13] proposed a method to iteratively estimate the blur kernel. Kligler et al. [2] introduced KernelGAN, which trains solely on the LR test image at test time, and learns its internal distribution of patches. Researchers also built complex models for image degradation, to augment the robustness of the SR model. Zhang et al. [51] designed a complex degradation model that consists of randomly shuffled blur, downsampling and noise degradations. Wang et al. [43] used a high-order degradation model to better simulate complex real-world degradations.

Zhang et al. [55] indicated that when the degradation distribution during both training and testing perfectly matches, the SR model exhibits favorable generalization and achieves high performance simultaneously. The binning method is utilized to adjust the joint distribution of the three parameters of a widely-used image degradation model, helping the training degradation distribution to better match the testing distribution. Both their work and our work aim to align degradation distributions to achieve good performance and generalization. However, they concentrate on aligning the distributions of degradation parameters within a given model, while we focus on developing a more suitable family of degradation functions.

**Adversarial Training.** Adversarial training improves the model robustness by training on adversarial examples generated by gradient-based method [12]. Madry et al. [32] studied the adversarial robustness of neural networks through the lens of robust optimization. Tramer et al. [41] proposed an ensemble adversarial training on adversarial examples generated from a number of pretrained models. Kolter and Wong [47] developed a provable robust model that minimizes worst-case loss over a convex outer region. Athalye et al. [1] demonstrated that adversarial training on PGD adversarial examples was to be the state-of-of-art defense model. Yue et al. [49] and Castillo et al. [4] both used adversarial examples during training to improve SR model's capacity to process noisy inputs, which is different with our method because they only considered adversarial noise rather than general real-world degradations.

**Domain Generalization.** Domain generalization aims to achieve out-of-distribution generalization by using only source data for model learning. Most existing approaches belong to the category of domain alignment [34], where the central idea is to minimize the difference among source domains for learning domain-invariant representations. Meta-learning [9] are also used to solve domain generalization by exposing a model to domain shift during training with a hope that the model can better deal with domain shift in unseen domains. In the context of domain generalization, the work most related to ours is RandConv [48]. It is based on the idea of using randomly initialized, single-layer convolutional neural network to transform the input images to novel domains. Since the weights are randomly sampled from a Gaussian distribution at each iteration and no learning is performed, the transformed images mainly contain random color distortions, which do not contain meaningful variations and are best to be mixed with the original images before passing to the task network.

**Identity Mapping in Deep Learning.** Identity mappings are widely used in deep learning methods, typically as network layers rather than degradation priors. Sun et al. [14] used identity mappings as the skip connections and after-addition activation, to make the training easier and improves

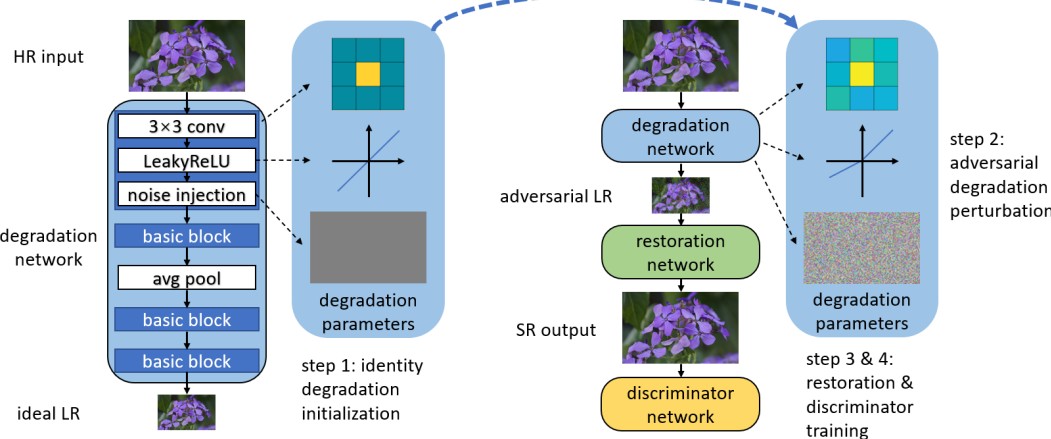

Figure 2: Illustration of the training procedure of our real-world super-resolution method with the proposed adversarial neural degradation model. Every single optimization step of the whole network can be divided into four sub-steps, and we highlight the internal state of the degradation network in the first two sub-steps.

model generalization. Zhang et al. [50] used an identity mapping task to study memorization and generalization of overparameterized networks in the extreme cases.

## 3    Adversarial Neural Degradation for Blind Super-Resolution

Before discussing our new robust real-world SR method, we would like to emphasize once again that the degradation prior employed in our approach is inspired by the following two observations of image degradations.

1. Almost all types of image degradation could find a corresponding operation in a standard convolutional neural network.
2. Almost all moderate image degradations could be considered as small deviations from the identity transformation.

Once we identify the commonalities among various image degradations, we can naturally propose a simple and elegant degradation prior that encompasses all of these degradations. That is, we take slightly deviated identity convolutional neural networks as prior for various real-world image degradations.

We illustrate the entire training procedure of our SR method with the proposed degradation prior in Fig. 2. The entire neural network can be divided into three parts: a degradation network, a restoration network, and an optional discriminator network. A single optimization step of the entire network can be further divided into the following four sub-steps. First, we initialize the degradation network to the identity transformation. Next, we adversarially search for a small deviation of the initialized degradation network that would cause the restoration network to perform the worst. Then, we optimize the restoration network based on the identified degradation case. Finally, we upgrade the discriminator network to distinguish restoration outputs from real images. We repeat the optimization steps of the entire network multiple times during training. Once the training is complete, we can discard the degradation network and the discriminator network, and only use the trained restoration network for inference.

In the following subsections, we will first describe the degradation network architecture and the reason why it can function as a prior to incorporate various image degradations. Then, we will explain the method used to initialize the degradation network to the identity transformation. Next, we will discuss the approach to perturb the degradation network, enabling it to represent complex image degradations. Subsequently, we will outline the adversarial training procedure of the SR model. Finally, we will analyze the training and inference efficiency of the method.

## 3.1 Degradation Network Architecture

Since we argue that almost all types of image degradation could find a corresponding operation in a standard convolutional neural network, the degradation network does not need much design for its architecture to work as a prior to include various image degradations. We can simply concatenate common convolutional neural network layers which could represent these image degradations.

Convolution layer, which is the most common layer type, is used in the degradation network to represent filter related degradations, like blur and ringing. These degradation types are also very common in the real world. Blur can be caused by camera movement or out of focus, and ringing can be caused by image compression or image sharpening technique.

Non-linear activation layer is used in the degradation network to represent global non-linear color changes, like color fading and posterization [35]. Color fading can be caused by inaccurate color response of old films, and posterization can be caused by color quantization in image compression.

Both convolution layer and activation layer can only represent spatially homogeneous image degradations, and their abilities are limited by the space invariant property of normal convolutional neural network. To represent spatially heterogeneous degradations like block artifacts in compressed images or dust spots in old images, we need to use a relatively less common layer called noise injection layer [19], which adds noise to its input in a pixel-wise manner. Combined with other layers, noise injection layer makes the degradation network able to represent complex spatially heterogeneous degradations.

Pooling layer is the last layer type we would like to discuss, and it can directly represent a downsampling process. We use anti-aliased average pooling layer [53] rather than a normal average pooling layer to avoid aliasing in the downsampling process.

For convolution, activation and noise layer, we would like to use multiple layers of the same type in the degradation network, to make the network able to represent complicated and higher order degradations. But only one pooling layer is used in the degradation network, since it is hard to break one pooling layer with integer downsampling factor down into multiple ones with non-integer factor. We combine a $3 \times 3$ convolution layer, a LeakyReLU activation layer [31] and a noise layer to form our basic block, put 5 basic blocks before and after an average pooling layer, and put a $3 \times 3$ convolution layer at the end. Number of channels in the degradation network are all 64, except for the input channels of the first convolution layer and the output channels of the last convolution layer, which are both 3 to take RGB images as input and output of the degradation network.

## 3.2 Identity Degradation Network Initialization

Due to overparameterization of neural networks, there are infinitely many parameter solutions to make a network represent the identity transformation, even if the network architecture is fixed [50]. However, two different parameter solutions, which could identically represent the same function, may have totally different behaviors of functions in their own parameter neighbourhood. We take slightly deviated identity neural networks as prior for various real-world image degradations. If all these networks are deviated from one or a few parameter solutions, their behaviors would be severely biased and cannot cover a variety of image degradations. Therefore, we need a fast initialization method to generate a lot of identity neural networks with different parameters.

The most straightforward initialization method, which trains networks on the identity mapping task by minimizing error using gradient descent, is way too slow for our SR model training. We propose a fast method that can initialize the degradation network to the identity transformation. Our method only takes a few small matrix multiplications and one singular value decomposition, while the most straightforward initialization method takes millions of training steps.

Before discussing our method to initialize the degradation network to the identity transformation, we would like to first clarify the meaning of the identity transformation in this work. For convolution, activation and noise layer in our degradation network, the definition of the identity transformation is strictly applicable, since the size of their input is the same as the size of their output. But for the pooling layer, the input feature is downsampled by a scale factor, so the strictly defined identity transformation no longer exists. In this work, we treat the ideal downsampling operation as a visually

identity transformation[2], and use the anti-aliased average pooling layer [53] as an approximation of the ideal downsampling. This interpretation is reasonable, because an image and its ideally downsampled counterpart are very similar from the perspective of the human visual system.

To make the degradation network represent the identity transformation, it must first be linear. So first we remove all nonlinearities in the network, by setting the negative slopes of all LeakyReLU activation layers [31] to one. And we set all additive noises in all noise injection layers to zero. An output pixel of an identity network can only be affected by the counterpart pixel of the input. So then we only fill the center slices of all convolution kernels with nontrivial values, and set all other values of all convolution kernels to zero. By this way, we essentially simulate $1 \times 1$ convolutions with $3 \times 3$ convolutions. We initialize the center slices of all but the last convolution kernels with Xavier Initialization [10] of $1 \times 1$ convolutional fan mode, to make all layers of the degradation network have a stable variance of responses. To finalize the identity network initialization, we merge all but the last $1 \times 1$ convolutions into one $1 \times 1$ convolution, by multiply center slices of convolution kernels as matrices. We compute the Moore-Penrose pseudoinverse of the merged matrix, and fill the result into the center slice of the last $3 \times 3$ convolution kernel.

We did not mention the anti-aliased average pooling layer in our initialization method. That is because it is a strictly defined linear operator, and we already claim it as a visually identity transformation. As long as the remaining part of the network is a strictly defined identity transformation, the whole network would be a visually identity transformation, or in another word, would be the ideal downsampling.

### 3.3 Adversarial Degradation Perturbation

Once we initialize the degradation network to the identity transformation, we are at the starting point to various image degradations. The next thing we need to do is to perturb the identity degradation network, and the slightly deviated identity degradation network would represent a real-world image degradation case. We can use the network to quickly generate abundant perfectly aligned HR and LR image pairs, by taking HR images as input of the degradation network and collecting the outputs. And finally, we can train a SR model with the collected HR and LR image pairs. Since the degradation prior includes various real-world image degradations, the SR model trained by this way could reconstruct various real-world degraded images well.

So how do we perturb the identity degradation network? The most straightforward way is to add small random numbers to all parameters of the degradation network, and that means to take a small random step from the original identity transformation on the degradation space. A lot of degradation networks which are independently perturbed by this way would cover a neighbourhood of the identity transformation on the degradation space. If we train a SR model with HR and LR image pairs constructed by many of those networks, the trained model would have a good average performance on the covered degradation set.

However, instead of having a good average performance for regular degradations, we want our real-world SR model to be as robust as possible. In other word, we want our SR model to have a good worst-case performance. That is because the real-world situation is always more complicated than laboratory situations. We want our real-world SR model to keep having a satisfactory performance, even for images might have been suffered from rare or unpredictable real-world degradations.

To achieve such a goal, we perturb the identity degradation network adversarially instead of randomly. That means, we adversarially search small perturbations on all parameters of the degradation network, which could make the SR model perform the worst on HR and LR image pairs constructed by the degradation network. During SR model training, we keep searching those worst cases dynamically, and keep optimizing the SR model based on the worst degradation case for the moment. By this way, the worst-case performance of the SR model would be gradually improved, and will finally converge to the robust model with the highest lower bound.

---

[2]More formally, a function $f$ is a visually identity transformation if for every image $X$, there exists a scale factor $s$ such that $f(X) = D(X; s)$, where $D$ is the ideal downsampling function.

### 3.4 Super-Resolution Model Training

To better show the advantage of the proposed neural degradation prior and the adversarial degradation training, we adopt the ESRGAN [44] as our SR model. The SR model training procedure solves the following optimization problem:

$$
\min_{\theta_G} \{ \, \mathbb{E}_{I^{HR}} [\max_{\theta_F \in S} L_{cont}(I^{HR}; \theta_G, \theta_F)]
$$
$$
+ \lambda \max_{\theta_D} \mathbb{E}_{I^{HR}} [\max_{\theta_F \in S} L_{GAN}(I^{HR}; \theta_G, \theta_D, \theta_F)] \}
\tag{1}
$$

where $S = \{\theta | \|\theta - \theta^{id}\|_2 < \varepsilon$, and $F_{\theta^{id}}$ is the identity transformation$\}$. $G$ and $D$ are generator (restoration network) and discriminator of the SR model respectively. $F$ is the degradation network. $\theta$ stands for parameter of network. $I^{HR}$ represents the high-resolution images. $L_{cont}$ and $L_{GAN}$ are the content loss and the GAN loss [22] respectively. $\lambda$ is the coefficient to balance the two loss terms. The content loss $L_{cont}$ is the sum of the 1-norm loss and the VGG loss [17]:

$$
L_{cont}(I^{HR}; \theta_G, \theta_F) = \|I^{HR} - G_{\theta_G}(F_{\theta_F}(I^{HR}))\|_1
$$
$$
+ \sum_j c_j \|\phi_j(I^{HR}) - \phi_j(G_{\theta_G}(F_{\theta_F}(I^{HR})))\|_2^2
\tag{2}
$$

where $\phi_j$ is the feature map of $j$th convolution layer of the VGG network [37], and $c_j$ is the coefficient for term of the $j$th layer. The GAN loss $L_{GAN}$ is:

$$
L_{GAN}(I^{HR}; \theta_G, \theta_D, \theta_F) = \log D_{\theta_D}(I^{HR})
$$
$$
- \log D_{\theta_D}(G_{\theta_G}(F_{\theta_F}(I^{HR})))
\tag{3}
$$

The optimization problem in Equation 1 is more complicated than the ordinary two-player minimax problem in previous GAN-based SR methods [22, 44, 43]. There are three networks involved in the optimization, and each network tries to compete or cooperate with other two networks. The restoration network (i.e., generator) takes LR output of the degradation network as its input, generates SR images with realistic and natural texture, and tries to fool the discriminator network into believing the generated SR images are actually natural HR images. The discriminator network tries to distinguish SR output of the restoration network from natural HR images. The degradation network takes natural HR images as its input, generates LR images with moderate but also complicated image degradations, tries to make the restoration network to generate unsatisfying results and to make it easier for the discriminator network to distinguish. The optimization procedure drives the restoration network to improve, until it is robust enough and can keep generating perceptually satisfying SR results, even if its LR input images are suffered from complicated real-world degradations.

### 3.5 Training and Inference Efficiency

Compared with previous SR methods, our SR model needs to cost more time during training phase. When we train our GAN-based SR method with adversarial neural degradation, we need to perturb the identity degradation network, optimize the restoration network and the discriminator network, in an alternating manner. The adversarial perturbation of the identity degradation network is done by taking gradient steps. In this work, before each operation step of the SR model, we use 5 gradient steps for adversarial degradation perturbation. That requires additionally 5 forward and backward passes through the whole network, including the degradation network and the SR model. Thankfully, the degradation network is much smaller than the SR model. That is reasonable because degradation is much easier than restoration, which is a general property for all inverse problems. So the degradation network itself does not cost much, most of the additional training cost is due to the adversarial training procedure. As we mentioned before, we adopt the ESRGAN as our SR model. It would cost 8.92 TFLOPs for one training step of ESRGAN on a training batch, while would cost 57.85 TFLOPs for our SR model training with the same training setting. So our method need an increase in training time of a factor of 6.49.

However, what is more important for our real-world SR method is inference efficiency. That is because once we finish the training, the robust SR model we get would not need retraining or finetune for

Table 1: Quantitative comparison with state-of-the-art methods on real-world blind image super-resolution benchmarks. The best and second best results are highlighted in red and blue, respectively.

| Method | RealSR(×4) [3] | | DRealSR(×4) [46] | | SupER(×4) [21] | | ImgPairs(×2) [18] | |
|---|---|---|---|---|---|---|---|---|
| | PSNR↑ | LPIPS↓ | PSNR↑ | LPIPS↓ | PSNR↑ | LPIPS↓ | PSNR↑ | LPIPS↓ |
| KernelGAN [2] | 25.13 | 0.3349 | 28.56 | 0.3978 | 25.65 | 0.3445 | 26.74 | 0.3340 |
| DAN [16] | 27.80 | 0.4114 | 30.59 | 0.4111 | 32.19 | 0.2064 | 28.56 | 0.2802 |
| BSRNet [51] | 27.35 | 0.3084 | 29.49 | 0.3411 | 32.11 | 0.2532 | 28.59 | 0.3915 |
| BSRGAN [51] | 26.51 | 0.2685 | 28.35 | 0.2929 | 29.18 | 0.2181 | 28.13 | 0.3346 |
| Real-ESRNet [43] | 26.79 | 0.2939 | 28.50 | 0.3257 | 30.89 | 0.2496 | 28.34 | 0.3858 |
| Real-ESRGAN [43] | 25.85 | 0.2728 | 27.92 | 0.2818 | 27.55 | 0.2046 | 28.12 | 0.3679 |
| SwinIR-Real [26] | 26.43 | 0.2515 | 28.29 | 0.2739 | 28.27 | 0.1889 | 28.11 | 0.3464 |
| DCLS [29] | 27.83 | 0.4080 | 28.32 | 0.4760 | 32.71 | 0.1985 | 28.64 | 0.2844 |
| PDM-SRGAN [28] | 21.96 | 0.3717 | 24.32 | 0.3668 | 25.31 | 0.2710 | 26.11 | 0.3788 |
| FeMaSR [5] | 25.42 | 0.2937 | 26.59 | 0.3374 | 25.45 | 0.2419 | 27.03 | 0.3400 |
| DASR [25] | 27.18 | 0.3113 | 29.72 | 0.2962 | 29.73 | 0.1476 | 28.34 | 0.3412 |
| ReDegNet [23] | 24.77 | 0.2800 | 26.24 | 0.2995 | 26.60 | 0.1785 | 27.06 | 0.3930 |
| ANDNet (ours) | 28.47 | 0.2599 | 30.97 | 0.3381 | 32.96 | 0.2125 | 28.75 | 0.2786 |
| ANDGAN (ours) | 26.34 | 0.2326 | 28.95 | 0.2610 | 29.85 | 0.1372 | 27.78 | 0.2598 |

unpredicted image degradations. Once the training is done, we can discard the degradation network and the discriminator network, and only use the trained restoration network for inference. That means, compared with previous SR methods, our SR method does not need additional computational and storage cost during inference phase.

## 4 Experiments

### 4.1 Datasets

Widely used datasets for SR evaluation, like Urban100 [15] and DIV2K [39], are not suitable for the study of real-world SR, because they only contain HR images and their LR counterpart need to be generated by bicubic downsampling. We perform experiments on four datasets constructed for real-world SR evaluation: RealSR [3], DRealSR [46], SupER [21], and ImagePairs [18]. RealSR and DRealSR are datasets containing HR and LR image pairs captured on the same scene by adjusting the focal length of digital cameras. SupER includes HR and LR image pairs constructed by camera hardware binning, which aggregates adjacent pixels on the sensor array. ImagePairs includes HR and LR image pairs captured by a HR camera and a LR camera, which are aligned and mounted on a rig with a beam splitter. These datesets are constructed in different ways, so they could provide a comprehensive evaluation for real-world SR methods.

### 4.2 Quantitative Metrics

We use four quantitative metrics for quality assessment of SR images: PSNR, SSIM [45], LPIPS [54], and NIQE [33]. PSNR and SSIM are calculated on Y channel of transformed YCbCr space for fair comparison [40]. They are more focused on low-level pixel-wise image differences, and they are metrics suitable for PSNR-oriented SR models. LPIPS is a learned metric for full-reference image quality assessment. We could use the preceding three full-reference metrics, since all datasets we used in the experiments have pixel-wise aligned HR and LR image pairs. Considering that GAN-based SR methods may generate detailed textures, which is although realistic but different from the ground truth, we also use NIQE, a no-reference metric for image quality evaluation. Both LPIPS and NIQE agree better with human visual perception, and they are metrics suitable for perceptual quality-oriented SR models.

### 4.3 Training Details

We use DIV2K [39], Flickr2K [27] and WED [30] as HR image datasets for training. The training HR patch size is set to 256 and the batch size is set to 48. Following BSRGAN [51] and Real-ESRGAN [43], we train two SR models with our method: a PSNR-oriented model noted as ANDNet,

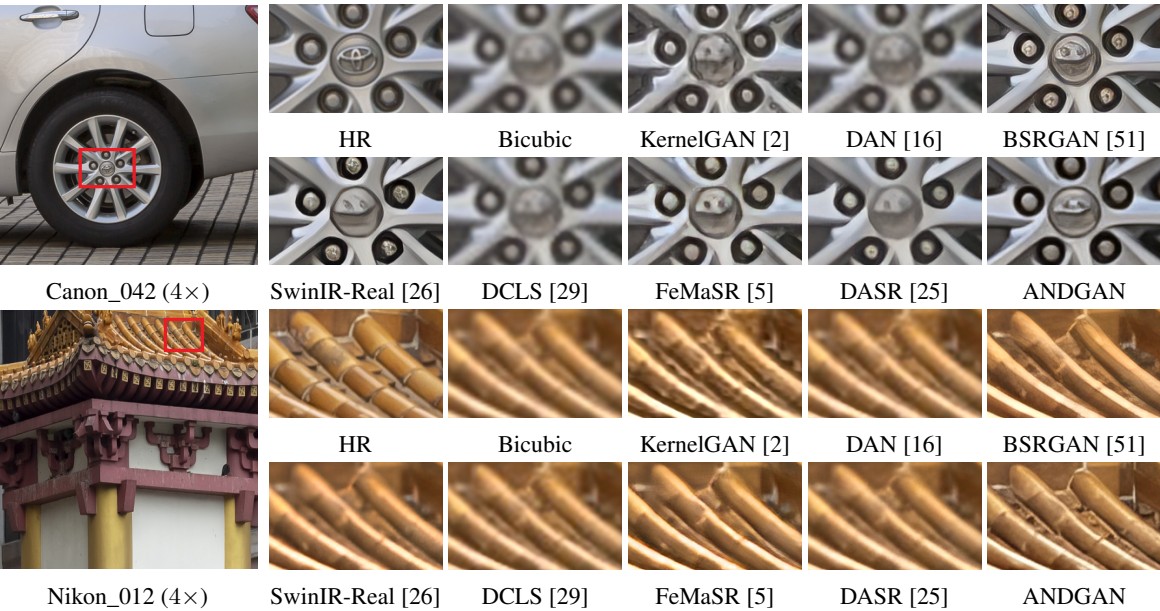

Figure 3: Qualitative comparisons on real-world images from RealSR [3] dataset with scale factor 4.

and a perceptual quality-oriented model noted as ANDGAN. First, we train ANDNet with the L1 loss only, for $1 \times 10^6$ iterations with $1 \times 10^{-4}$ learning rate. Then we use the trained ANDNet as an initialization for the generator of the ANDGAN, and train the whole ANDGAN model with both the content loss and the GAN loss in Equation 1, which are balanced by $\lambda = 0.1$, for $5 \times 10^5$ iterations with $1 \times 10^{-4}$ learning rate. We use Adam optimizer [20] for both generator and discriminator training.

We use projected gradient descent method [32] to adversarially search a small deviation of the identity degradation network. Before every training step for the restoration network, we initialize the degradation network to the identity transformation, and run 5 iterations of projected gradient descent with step size of 6 and perturbation size $\varepsilon = 20$. We also need to balance the weights for convolutional degradation, noise and nonlinearity, to make their respective intensity close to real-world degradation. When we calculate the L2 norm $\|\theta - \theta^{id}\|_2$ to determine the perturbation set $S$ in Equation 1, we use scale factors of 1, 10, 50 for term of convolutional degradation, noise and nonlinearity respectively. Note that a larger scale factor means a stronger suppression for the degradation type.

## 4.4 Comparisons with Prior Works

We compare both our PSNR-oriented model and the perceptual quality-oriented model with several state-of-the-art methods. Quantitative results are shown in Table 1, and visual comparison between different methods are shown in Fig. 3. Due to limited space, we can only provide most relevant results here. More detailed results can be found in the supplementary material.

## 4.5 Ablation Study

In order to study the effects of each component in the proposed blind SR method, we gradually modify the AND model and compare their quantitative performances. The comparisons are shown in Table 2. For cases where neural degradation is removed, we either use an additive noise model for adversarial training or employ bicubic downsampling for non-adversarial training. In situations where identity initialization is eliminated, we randomly initialize all $3 \times 3$ convolution kernels in the degradation network using Xavier Initialization. We can observe that all three major components, namely adversarial perturbation, neural degradation, and identity initialization, are necessary.

The first column of Table 2, labeled as "Configuration", is a name and an explanation for a particular ablation setting. If we do not use adversarial perturbation and neural degradation at all, our method

Table 2: Comparisons showing the effects of each component in the AND model, tested on the RealSR [3] dataset with a scale factor of 4.

| Configuration | Adversarial Perturbation | Neural Degradation | Identity Initialization | PSNR↑ of ANDNet | LPIPS↓ of ANDGAN |
|---|---|---|---|---|---|
| Classical SR training | ✗ | ✗ | - | 26.53 | 0.4245 |
| Adversarial noise training | ✓ | ✗ | - | 26.60 | 0.4194 |
| Synthetic data augmentation | ✗ | ✓ | ✓ | 27.31 | 0.3089 |
| Severe random style shift | ✓ | ✓ | ✗ | 11.84 | - |
| Complete AND model | ✓ | ✓ | ✓ | 28.47 | 0.2326 |

would become a classical SR training method [8, 27, 56], which assumes that the image degradation model is an ideal bicubic downsampling. If we retain solely the adversarial perturbation without inducing neural degradation, it implies the utilization of an adversarial noise training method similar to [4, 49]. If we employ neural degradation with identity initialization but without adversarial perturbation, our method becomes similar to SR training with synthetic data augmentation [51, 43]. If we only remove the identity initialization from our method, the generated LR patches would no longer be visually similar to the HR patches. While the skeleton of the LR patches would remain unaffected, their color and texture would undergo a dramatic change [48]. We can observe that all three major components, namely adversarial perturbation, neural degradation, and identity initialization, are necessary.

## 5   Conclusions

We propose a neural degradation prior that encompasses various image degradations in the real world. Specifically, an untrained convolutional neural network, which deviates slightly from the identity transformation, can serve as a prior for various real-world image degradations. We employ adversarial searches to find small deviations in the degradation network during the training of the SR model. This approach allows the restoration model to continuously optimize itself on the worst degradation case, thus achieving robustness.

## Acknowledgement

This work is supported by the Natural Sciences and Engineering Research Council of Canada.

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
