# A Further Explanation of the Training Procedure

Compared with previous GAN-based SR methods [11, 24, 23], our training procedure has two distinct parts: identity degradation initialization and adversarial degradation perturbation. In this section, we provide a detailed explanation of these two components.

## A.1 Identity Degradation Initialization

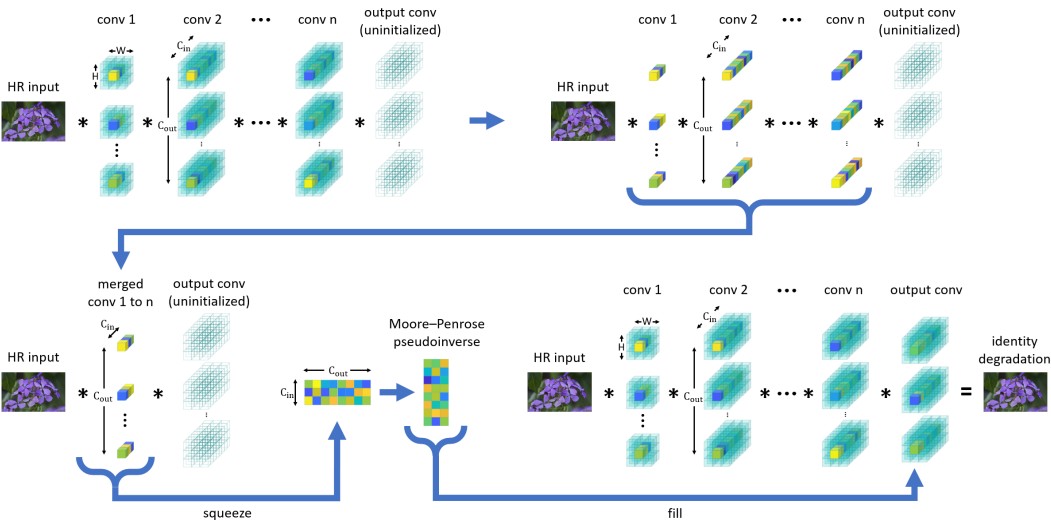

Figure 1: Illustration of the identity degradation initialization method in our training procedure. Only the convolution layers in the degradation network are shown in the figure.

The process of identity degradation initialization is illustrated in Fig. 1. Please note that in each initialization procedure, we first set the negative slopes of all LeakyReLU layers [20] to one and set additive noises in all noise injection layers [8] to zero. Thus, these two types of layers can be removed from the degradation network and are not shown in Fig.1. Then we initialize the center slices of all but the last convolution kernels (conv 1 to n) with Xavier Initialization [4] of $1 \times 1$ convolutional fan mode. We set all other values of these kernels to zero and leave the last convolution kernel (output conv) uninitialized. We can simplify these initialized $3 \times 3$ convolutions (conv 1 to n) to $1 \times 1$ convolutions and merge them into one $1 \times 1$ convolution, taking advantage of the associativity of convolution. Finally, we squeeze the merged $1 \times 1$ convolution filter (4-D tensor) into a 2-D matrix of the same size, compute the Moore-Penrose pseudoinverse of the merged matrix, fill the result into the center slice of the last $3 \times 3$ convolution kernel (output conv), and set all other values of the kernel to zero. This way, each output pixel of the degradation network is only affected by the counterpart pixel of the HR input. Computing the output pixel is equivalent to performing a matrix multiplication of the input pixel, a randomly initialized matrix, and its pseudoinverse. Thus, the property of the pseudoinverse guarantees that the entire degradation network is initialized to the identity transformation.

Please note that the anti-aliased average pooling layer [30] is not shown in Fig. 1. Since it is a strictly defined linear operator and we claim it as a visually identity transformation, as long as the remaining part of the network is a strictly defined identity transformation, the entire network would be a visually identity transformation or, in other words, the ideal downsampling.

Due to overparameterization of neural networks, there are infinitely many parameter solutions to make the degradation network represent the identity transformation. Our initialization method only takes $n - 1$ matrix multiplications, and one Moore-Penrose pseudoinverse by using the singular value decomposition (SVD). Our method strikes a nice balance between randomness in the network neighborhood and initialization speed.

---

**Algorithm 1** The general algorithm for AND training

---

**Require:** epoch number $N$, batch size $m$, step size $\alpha$, perturbation bound $\varepsilon$, perturbation steps $K$, learning rate $\eta$

**Require:** initial generator parameters $\theta_G$, initial discriminator parameters $\theta_D$.

    **for** $epoch = 1$ to $N$ **do**

        Initialize $\theta_F$ which makes the degradation network $F$ represent the identity transformation.

        Initialize perturbation on parameters of the degradation network $\delta \leftarrow 0$

        Sample a minibatch $\{x^i\}_{i=1}^m$ from the high-resolution images $I^{HR}$.

        **for** $k = 1$ to $K$ **do**

            $g_F \leftarrow \nabla_{\theta_F}[\frac{1}{m}\sum_{i=1}^m (L_{cont}(x^i; \theta_G, \theta_F + \delta) + \lambda L_{GAN}(x^i; \theta_G, \theta_D, \theta_F + \delta))]$

            $\delta \leftarrow \delta + \alpha \frac{g_F}{\|g_F\|_2}$

            **if** $\|\delta\|_2 > \varepsilon$ **then**

                $\delta \leftarrow \varepsilon \frac{\delta}{\|\delta\|_2}$

            **end if**

        **end for**

        $\theta_F \leftarrow \theta_F + \delta$

        $g_G \leftarrow \nabla_{\theta_G}[\frac{1}{m}\sum_{i=1}^m (L_{cont}(x^i; \theta_G, \theta_F) + \lambda L_{GAN}(x^i; \theta_G, \theta_D, \theta_F))]$

        $\theta_G \leftarrow$ Adam$(-g_G, \theta_G, \eta)$

        $g_D \leftarrow \nabla_{\theta_D}[\frac{1}{m}\sum_{i=1}^m \lambda L_{GAN}(x^i; \theta_G, \theta_D, \theta_F)]$

        $\theta_D \leftarrow$ Adam$(g_D, \theta_D, \eta)$

    **end for**

---

## A.2 Adversarial Degradation Perturbation and Training

The purpose of the entire training procedure is to solve the optimization problem proposed in Section 3.4, which we restate here for convenience:

$$\min_{\theta_G} \{ \mathbb{E}_{I^{HR}}[\max_{\theta_F \in S} L_{cont}(I^{HR}; \theta_G, \theta_F)]$$
$$+ \lambda \max_{\theta_D} \mathbb{E}_{I^{HR}}[\max_{\theta_F \in S} L_{GAN}(I^{HR}; \theta_G, \theta_D, \theta_F)]\} \tag{1}$$

where $S = \{\theta | \|\theta - \theta^{id}\|_2 < \varepsilon$, and $F_{\theta^{id}}$ is the identity transformation$\}$. $G$ and $D$ are generator (restoration network) and discriminator of the SR model respectively. $F$ is the degradation network. $\theta$ stands for parameter of network. $I^{HR}$ represents the high-resolution images. $L_{cont}$ and $L_{GAN}$ are the content loss and the GAN loss [11] respectively. $\lambda$ is the coefficient to balance the two loss terms.

We present the general algorithm for AND training, which includes the adversarial degradation perturbation method, in Algorithm 1. The algorithm is more complicated than the training algorithms of previous GAN-based SR methods [11, 24, 23], because there are not two, but three players in the minimax problem, i.e., the degradation network $F$, the generator $G$, and the discriminator $D$. For each single optimization step of the entire network, we first initialize the degradation network to the identity transformation. Next, we adversarially perturb the degradation network within a small neighborhood. The degradation network takes HR images as input and generates LR images with moderate yet complex image degradations. The restoration network, also known as the generator, aims to restore SR images from the degraded LR images. The adversarial degradation perturbation is designed to cause the restoration network to produce unsatisfactory results, characterized by low PSNR and easy distinguishability as fake images by the discriminator network. This adversarial degradation perturbation is accomplished through $K = 5$ projected perturbation steps [21]. Finally, we optimize the restoration network and the discriminator network using the adversarial LR images. We repeat the optimization steps of the entire network multiple times during training until the restoration network becomes robust enough to generate perceptually satisfying SR results, even when the LR input images are affected by complex degradations.

## B  Limitations

There is no single SR model that can handle every possible image degradation. This is a simple deduction drawn from the "no free lunch" theorem [27], and our method is certainly not an exception. Our SR method relies on the proposed neural degradation prior, which is inspired by the commonalities observed in various image degradations. Therefore, naturally, our SR model cannot effectively deal with a specific image degradation that deviates significantly from the two summarized commonalities.

When an image degradation introduces artifacts containing strong structures that are not accounted for in the neural degradation prior, our SR model struggles to handle the degradation. An illustrative degradation example is extreme JPEG compression [32], which produces severe block artifacts characterized by strong spatial structures in $8 \times 8$ blocks. While networks trained specifically for this task can utilize such structures, they are not explicitly captured in our proposed neural degradation prior. As a result, our SR model would not outperform an expert network in this case. Another similar degradation example is halftoning [9]. Halftone printing is a technique that uses ink dots of different sizes to simulate different grayscale levels, and it is used in old publications such as newspapers or books. Since the positions of the ink dots of a halftone image always have a very strong spatial pattern, which is not included in the neural degradation prior, our method cannot restore the halftone image well compared to an expert network.

The prior assumes that image degradations can be viewed as small deviations from the identity transformation. If this assumption fails for a specific image degradation, our SR model cannot handle the degradation well. A notable degradation example is the conversion of truecolor images to grayscale images [6]. This particular degradation is not a small deviation from the identity transformation, and thus, our SR model would not be able to colorize the input grayscale images in such cases. For the same reason, our method is also not suitable for directly enhancing low-light images [17]. Since the image degradations used in our SR training are slightly deviated from identity degradation networks, our trained model would not automatically adjust the image brightness, contrast, and color.

For certain image degradations, both assumptions in our neural degradation prior can fail simultaneously, representing the most challenging cases for our method. An example of such a case is image degradation in the image inpainting task [16]. In this task, the missing pixels often exhibit strong spatial structures, such as forming holes and stripes on the image. Moreover, the signal of these pixels is entirely absent, rather than being a small deviation from the identity transformation. Consequently, our image restoration method is unable to address this type of degradation.

## C  More Experimental Results

We utilize four quantitative metrics to assess the quality of SR images: PSNR, SSIM [25], LPIPS [31], and NIQE [22]. PSNR and SSIM primarily focus on low-level pixel-wise image differences, making them suitable metrics for PSNR-oriented SR models. On the other hand, LPIPS and NIQE align better with human visual perception, making them more suitable for perceptual quality-oriented SR models. Due to limited space, we only present PSNR and LPIPS results in Section 4.4. Here we provide the remaining SSIM and NIQE results in Table 1. Additionally, we offer visual comparisons with state-of-the-art methods in Fig.2. The LR images in Fig. 2 are obtained from the DRealSR [26] dataset.

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

Table 1: Quantitative comparison with state-of-the-art methods on real-world blind image super-resolution benchmarks. The best and second best results are highlighted in red and blue, respectively.

| Method | RealSR(×4) [2] | | DRealSR(×4) [26] | | SupER(×4) [10] | | ImgPairs(×2) [7] | |
|---|---|---|---|---|---|---|---|---|
| | SSIM↑ | NIQE↓ | SSIM↑ | NIQE↓ | SSIM↑ | NIQE↓ | SSIM↑ | NIQE↓ |
| KernelGAN [1] | 0.7407 | 6.946 | 0.8314 | 8.550 | 0.7831 | 6.844 | 0.7467 | 6.291 |
| DAN [5] | 0.7882 | 8.099 | 0.8608 | 9.137 | 0.8880 | 5.886 | 0.7917 | 5.692 |
| BSRNet [29] | 0.8076 | 7.271 | 0.8587 | 8.060 | 0.8800 | 6.322 | 0.8311 | 6.391 |
| BSRGAN [29] | 0.7750 | 4.650 | 0.8205 | 4.681 | 0.8292 | 4.549 | 0.8152 | 5.450 |
| Real-ESRNet [23] | 0.8067 | 7.142 | 0.8484 | 7.829 | 0.8563 | 6.408 | 0.8264 | 6.026 |
| Real-ESRGAN [23] | 0.7735 | 4.676 | 0.8247 | 4.716 | 0.8082 | 3.944 | 0.8192 | 4.812 |
| SwinIR-Real [14] | 0.7865 | 4.678 | 0.8272 | 4.665 | 0.8360 | 3.776 | 0.8133 | 4.315 |
| DCLS [19] | 0.7892 | 8.023 | 0.8188 | 9.273 | 0.8927 | 5.892 | 0.7962 | 5.773 |
| PDM-SRGAN [18] | 0.6815 | 6.798 | 0.7728 | 7.518 | 0.8048 | 5.015 | 0.7788 | 5.316 |
| FeMaSR [3] | 0.7531 | 4.737 | 0.7683 | 4.218 | 0.7429 | 4.873 | 0.7477 | 4.719 |
| DASR [13] | 0.7867 | 5.969 | 0.8543 | 6.347 | 0.8508 | 3.881 | 0.8204 | 4.830 |
| ReDegNet [12] | 0.7754 | 5.049 | 0.8124 | 4.685 | 0.8305 | 3.993 | 0.8130 | 5.368 |
| ANDNet (ours) | 0.8232 | 7.541 | 0.8773 | 8.360 | 0.9004 | 6.302 | 0.8415 | 5.514 |
| ANDGAN (ours) | 0.7854 | 4.018 | 0.8244 | 4.045 | 0.8579 | 3.493 | 0.7858 | 3.748 |

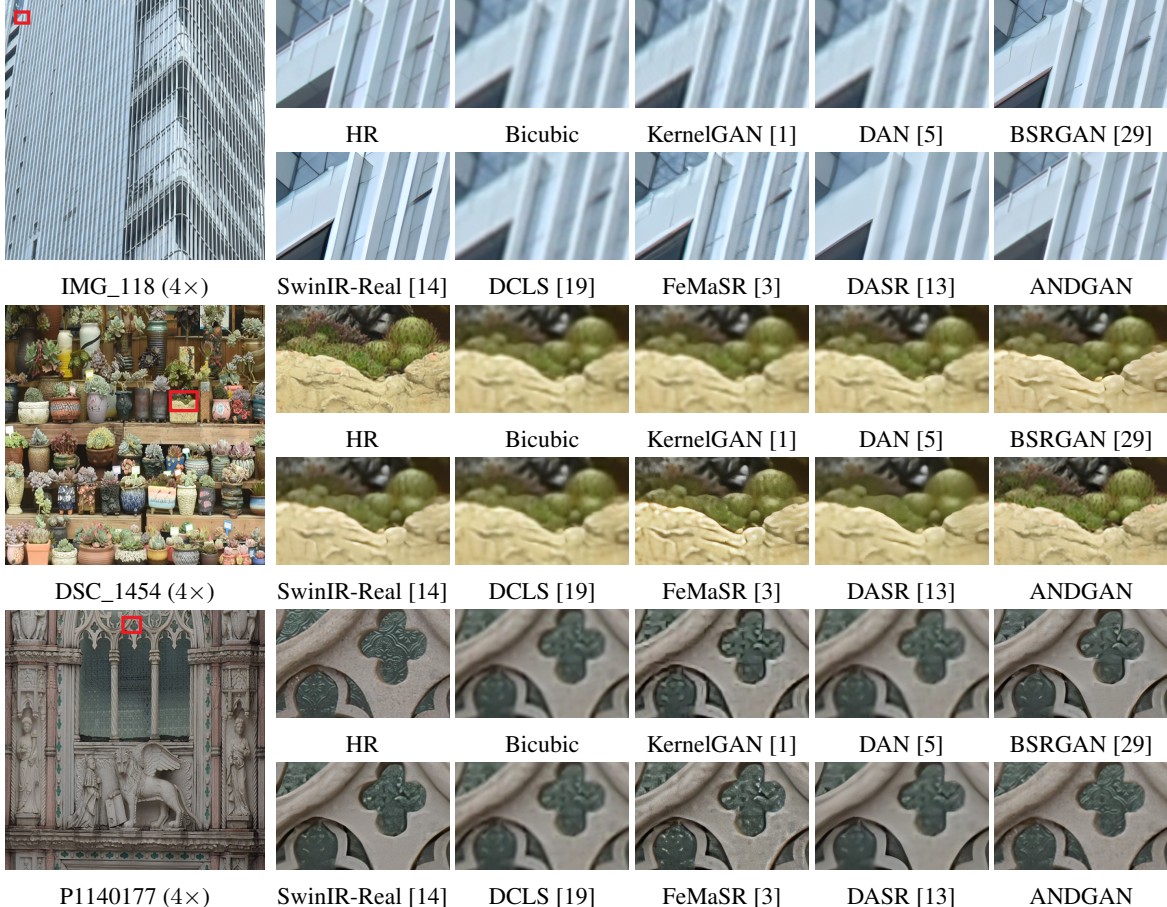

Figure 2: Qualitative comparisons on real-world images from DRealSR [26] dataset with scale factor of 4.

[4] Xavier Glorot and Yoshua Bengio. Understanding the difficulty of training deep feedforward neural networks. In *Proceedings of the thirteenth international conference on artificial intelligence and statistics*, pages 249–256. JMLR Workshop and Conference Proceedings, 2010.

[5] Yan Huang, Shang Li, Liang Wang, Tieniu Tan, et al. Unfolding the alternating optimization for blind super resolution. *Advances in Neural Information Processing Systems*, 33:5632–5643, 2020.

[6] Satoshi Iizuka, Edgar Simo-Serra, and Hiroshi Ishikawa. Let there be color! joint end-to-end learning of global and local image priors for automatic image colorization with simultaneous classification. *ACM Transactions on Graphics (ToG)*, 35(4):1–11, 2016.

[7] Hamid Reza Vaezi Joze, Ilya Zharkov, Karlton Powell, Carl Ringler, Luming Liang, Andy Roulston, Moshe Lutz, and Vivek Pradeep. Imagepairs: Realistic super resolution dataset via beam splitter camera rig. In *Proceedings of the IEEE/CVF Conference on Computer Vision and Pattern Recognition Workshops*, pages 518–519, 2020.

[8] Tero Karras, Samuli Laine, and Timo Aila. A style-based generator architecture for generative adversarial networks. In *Proceedings of the IEEE/CVF conference on computer vision and pattern recognition*, pages 4401–4410, 2019.

[9] Tae-Hoon Kim and Sang Il Park. Deep context-aware descreening and rescreening of halftone images. *ACM Transactions on Graphics (TOG)*, 37(4):1–12, 2018.

[10] Thomas Köhler, Michel Bätz, Farzad Naderi, André Kaup, Andreas Maier, and Christian Riess. Toward bridging the simulated-to-real gap: Benchmarking super-resolution on real data. *IEEE transactions on pattern analysis and machine intelligence*, 42(11):2944–2959, 2019.

[11] Christian Ledig, Lucas Theis, Ferenc Huszár, Jose Caballero, Andrew Cunningham, Alejandro Acosta, Andrew Aitken, Alykhan Tejani, Johannes Totz, Zehan Wang, et al. Photo-realistic single image super-resolution using a generative adversarial network. In *Proceedings of the IEEE conference on computer vision and pattern recognition*, pages 4681–4690, 2017.

[12] Xiaoming Li, Chaofeng Chen, Xianhui Lin, Wangmeng Zuo, and Lei Zhang. From face to natural image: Learning real degradation for blind image super-resolution. In *Computer Vision–ECCV 2022: 17th European Conference, Tel Aviv, Israel, October 23–27, 2022, Proceedings, Part XVIII*, pages 376–392. Springer, 2022.

[13] Jie Liang, Hui Zeng, and Lei Zhang. Efficient and degradation-adaptive network for real-world image super-resolution. In *Computer Vision–ECCV 2022: 17th European Conference, Tel Aviv, Israel, October 23–27, 2022, Proceedings, Part XVIII*, pages 574–591. Springer, 2022.

[14] Jingyun Liang, Jiezhang Cao, Guolei Sun, Kai Zhang, Luc Van Gool, and Radu Timofte. Swinir: Image restoration using swin transformer. In *Proceedings of the IEEE/CVF international conference on computer vision*, pages 1833–1844, 2021.

[15] Bee Lim, Sanghyun Son, Heewon Kim, Seungjun Nah, and Kyoung Mu Lee. Enhanced deep residual networks for single image super-resolution. In *Proceedings of the IEEE conference on computer vision and pattern recognition workshops*, pages 136–144, 2017.

[16] Guilin Liu, Fitsum A Reda, Kevin J Shih, Ting-Chun Wang, Andrew Tao, and Bryan Catanzaro. Image inpainting for irregular holes using partial convolutions. In *Proceedings of the European conference on computer vision (ECCV)*, pages 85–100, 2018.

[17] Kin Gwn Lore, Adedotun Akintayo, and Soumik Sarkar. Llnet: A deep autoencoder approach to natural low-light image enhancement. *Pattern Recognition*, 61:650–662, 2017.

[18] Zhengxiong Luo, Yan Huang, Shang Li, Liang Wang, and Tieniu Tan. Learning the degradation distribution for blind image super-resolution. In *Proceedings of the IEEE/CVF Conference on Computer Vision and Pattern Recognition*, pages 6063–6072, 2022.

[19] Ziwei Luo, Haibin Huang, Lei Yu, Youwei Li, Haoqiang Fan, and Shuaicheng Liu. Deep constrained least squares for blind image super-resolution. In *Proceedings of the IEEE/CVF Conference on Computer Vision and Pattern Recognition*, pages 17642–17652, 2022.

[20] Andrew L Maas, Awni Y Hannun, Andrew Y Ng, et al. Rectifier nonlinearities improve neural network acoustic models. In *Proc. icml*, volume 30, page 3. Atlanta, Georgia, USA, 2013.

[21] Aleksander Madry, Aleksandar Makelov, Ludwig Schmidt, Dimitris Tsipras, and Adrian Vladu. Towards deep learning models resistant to adversarial attacks. *arXiv preprint arXiv:1706.06083*, 2017.

[22] Anish Mittal, Rajiv Soundararajan, and Alan C Bovik. Making a "completely blind" image quality analyzer. *IEEE Signal processing letters*, 20(3):209–212, 2012.

[23] Xintao Wang, Liangbin Xie, Chao Dong, and Ying Shan. Real-esrgan: Training real-world blind super-resolution with pure synthetic data. In *Proceedings of the IEEE/CVF International Conference on Computer Vision*, pages 1905–1914, 2021.

[24] Xintao Wang, Ke Yu, Shixiang Wu, Jinjin Gu, Yihao Liu, Chao Dong, Yu Qiao, and Chen Change Loy. Esrgan: Enhanced super-resolution generative adversarial networks. In *Proceedings of the European conference on computer vision (ECCV) workshops*, pages 0–0, 2018.

[25] Zhou Wang, Alan C Bovik, Hamid R Sheikh, and Eero P Simoncelli. Image quality assessment: from error visibility to structural similarity. *IEEE transactions on image processing*, 13(4):600–612, 2004.

[26] Pengxu Wei, Ziwei Xie, Hannan Lu, Zongyuan Zhan, Qixiang Ye, Wangmeng Zuo, and Liang Lin. Component divide-and-conquer for real-world image super-resolution. In *Computer Vision–ECCV 2020: 16th European Conference, Glasgow, UK, August 23–28, 2020, Proceedings, Part VIII 16*, pages 101–117. Springer, 2020.

[27] David H Wolpert and William G Macready. No free lunch theorems for optimization. *IEEE transactions on evolutionary computation*, 1(1):67–82, 1997.

[28] Chiyuan Zhang, Samy Bengio, Moritz Hardt, Michael C Mozer, and Yoram Singer. Identity crisis: Memorization and generalization under extreme overparameterization. *arXiv preprint arXiv:1902.04698*, 2019.

[29] Kai Zhang, Jingyun Liang, Luc Van Gool, and Radu Timofte. Designing a practical degradation model for deep blind image super-resolution. In *Proceedings of the IEEE/CVF International Conference on Computer Vision*, pages 4791–4800, 2021.

[30] Richard Zhang. Making convolutional networks shift-invariant again. In *International conference on machine learning*, pages 7324–7334. PMLR, 2019.

[31] Richard Zhang, Phillip Isola, Alexei A Efros, Eli Shechtman, and Oliver Wang. The unreasonable effectiveness of deep features as a perceptual metric. In *Proceedings of the IEEE conference on computer vision and pattern recognition*, pages 586–595, 2018.

[32] Xiaoshuai Zhang, Wenhan Yang, Yueyu Hu, and Jiaying Liu. Dmcnn: Dual-domain multi-scale convolutional neural network for compression artifacts removal. In *2018 25th IEEE International Conference on Image Processing (ICIP)*, pages 390–394. IEEE, 2018.