# OpenReview forum: "AND: Adversarial Neural Degradation for Learning Blind Image Super-Resolution"
_NeurIPS.cc/2023/Conference — NeurIPS 2023 poster_

### Official Review · Reviewer_BEQG · 2023-06-20

**Soundness:** 3 good
**Presentation:** 2 fair
**Contribution:** 2 fair
**Rating:** 3
**Confidence:** 5

**Summary:**

This paper proposes a method of AND to learn neural degradation for the task of blind image super-resolution. The core idea is learning to degrade HR images by neural networks, trying to synthesize real-world degradations. Based on the synthesized data, a restoration model can be well trained. The proposed AND model has a unique advantage over the current state of the art in that it can generalize much better to unseen degradation variants. Experimental studies on public datasets show its effectiveness.

**Strengths:**

+ The paper is generally well written, and it is easy to follow.

+ The paper conducts extensive studies including ablation studies, showing effective results.

**Weaknesses:**

There are several issues in this paper, including:

- The idea of learning degradation to better learn blind image super-resolution is not new, and has been studied in existing works, like "To learn image super-resolution, use a GAN to learn how to do image degradation first" (ECCV2018). The contribution can hardly be important.

- The claim "But regrettably, adversarial learning has not been applied to neural network models of signal restoration" is not valid. As a fact, there have been many works in this direction. For example, "Robust Real-World Image Super-Resolution against Adversarial Attacks"(ACMMM2021). There are more that are not given as examples.

- The experimental results need more clarification and explanations.

- Minor: L238, "By this way", L278, "So our method need an ..."


**Questions:**

1, the qualitative results are not promising and are still vague, compared with other methods. This needs more explanations.

**Limitations:**

The authors did not address the limitations.

---

> ### Author Rebuttal · Authors · 2023-08-10
>
> **Q1:** The idea of learning degradation to better learn blind image super-resolution is not new, and has been studied in existing works, like "To learn image super-resolution, use a GAN to learn how to do image degradation first" (ECCV2018).
>
> **A1:** You seem to misunderstand the main contribution of our paper. It may help for us to clarify further.
>
> The proposed neural degradation prior is an untrained neural network, which does not learn to imitate any specific degradation effects, hence not requiring degraded LR training images at all.  The untrained network is able to work as an image degradation prior, because it is designed to satisfy the following two common properties of a wide range of image degradations. First, the degradation effects can be modeled by CNN operations. Second, most degradations of interest can be considered as a small deviation from the identity transformation.
>
> On the other hand, the ECCV2018 paper uses a trained network to imitate the degradation effects using a given LR image set; the learnt degradation is then used to train the super-resolution model. This is a totally different methodology from ours, and the methodology difference leads to the inferior  generalization ability of the former to the latter. Their method would fail easily if the degradation of the chosen LR training images mismatches the real degradation at the inference stage, while our method can generalize much better to unseen degradation variants with the untrained neural degradation prior.
>
> We would like to show the generalization ability gap by experiments. However, the ECCV2018 paper focuses only on facial image SR, which is by itself a weakness, so we cannot compare it fairly with our method. Luckily, there is another paper published in ECCV2022, named "From face to natural image: Learning real degradation for blind image super-resolution". This ECCV2022 paper, like the ECCV2018 paper, also uses GAN to learn degradation from an LR image set but it can be applied to natural image SR task, so we can compare its performance with our method. The name of the ECCV2022 method is ReDegNet, and its performance is already shown in Table 1 of our paper (page 8) and in Table 1 of our supplementary material (page 4). The quantitative comparison proves that our method does generalize better on all four real-world SR datasets.
> ***
>
> **Q2:** The claim "But regrettably, adversarial learning has not been applied to neural network models of signal restoration" is not valid. As a fact, there have been many works in this direction. For example, "Robust Real-World Image Super-Resolution against Adversarial Attacks" (ACMMM2021).
>
> **A2:** Thank you for pointing this out. We will tune down our statement accordingly, and discuss the contribution of the ACMMM2021 paper in the revised version of our paper. However, here we would like to emphasize the difference between the ACMMM2021 paper and our paper.
>
> The ACMMM2021 paper borrows the earlier research in adversarial training for image classification tasks. But it does not account for the differences between the classification and restoration tasks. In the previous research on adversarial training for image classification tasks, Out-of-Distribution (OOD) perturbations are introduced through the deliberate efforts of malicious attackers. This approach utilizes pixelwise additive high-frequency noise as a concealed and effective perturbation attack. Note that the adversarial attack is very different from the type of signal degradations in restoration tasks. Specifically, the perturbations in super-resolution tasks encompass a mixture of blur, noise, and nonlinear transformations. As a result, for restoration tasks noise no longer predominantly influences the OOD perturbation as in classification tasks. Therefore, the proposed neural degradation prior is a more suitable perturbation model for real-world image restoration.
>
> The experimental settings of the ACMMM2021 paper and our paper also differ due to different understandings of real-world image SR tasks. In the ACMMM2021 paper's experiment, real-world LR images were not used directly as input for the SR model. Instead, they were first manipulated by a malicious attacker, which may not represent the most crucial scenario for real-world image SR. Conversely, our approach utilizes real-world LR images directly as input and demonstrates promising performance on real-world image SR datasets.
>
> ***
> **Q3:** The qualitative results are more vague, compared with other methods.
>
> **A3:** Yes, your observation is accurate. However, we must emphasize that the enhanced sharpness in the image outputs from other methods results from an artifact known as "watercolor-like artifacts" or "painterly artifacts". This artifact can occur when SR algorithms, especially GAN-based methods, generate images that exhibit characteristics similar to brush strokes or painterly textures, rather than natural or realistic details. The boundaries between two brush strokes would be very sharp, even sharper than the HR images. It is considered an artifact because it deviates from the original intention of SR, which is to enhance the detail and clarity of the image without introducing artistic or stylized characteristics. Compared to other methods, our SR output more closely resembles a natural HR image.
>
> ***
> **Q4:** The authors did not address the limitations.
>
> **A4:** We did discuss the limitations of our method in the Limitations section of our supplementary material (page 3).
>
> ***
> Since there are several misunderstandings regarding our paper, we would greatly appreciate it if you could review it again. If you have any further questions, please don't hesitate to reach out. We are more than willing to provide clarifications both within our discussion and in the paper itself.

---

> > ### Comment · Reviewer_BEQG · 2023-08-16
> > **Thanks for the rebuttal**
> >
> > Thanks a lot for providing the rebuttal. I appreciate the effort, but it fails to resolve my concerns. I think I did not misunderstand the paper. The paper indeed shares the core idea with the ECCV2018 paper, regardless of the trivial aspects. Some of my other concerns are neither resolved by the rebuttal. My initial rating will be kept.

---

> ### Author Response · Authors · 2023-08-16
> **Followup Response to Reviewer BEQG**
>
> Dear Reviewer BEQG,
>
> We would like to thank you again for the valuable time you devoted to reviewing our paper. We believe that we have addressed your concerns. Since the end of discussion period is getting close and we have not heard back from you yet, we would appreciate it if you kindly let us know of any other concerns you may have, and if we can be of any further assistance in clarifying them.
>
> Thank you once again for your contribution to our paper's development.
>
> Authors

---

### Official Review · Reviewer_Qwb9 · 2023-07-02

**Soundness:** 3 good
**Presentation:** 2 fair
**Contribution:** 3 good
**Rating:** 4
**Confidence:** 5

**Summary:**

This paper proposes an adversarial approach for blind image super-resolution. Instead of using combinations of synthetic degradations (e.g., Gaussian blur, JPEG compression), this paper proposes to use a degradation network to construct LR patches from HR patches during training. The degradation network is optimized with a constraint that the network parameters fall within a local region of that of an identity mapping. The proposed approach outperforms existing works on common datasets.

**Strengths:**

1. The proposed ANDNet and ANDGAN achieve promising performance on multiple common datasets, outperforming existing works.
2. The idea of using a degradation network is interesting.


**Weaknesses:**

1. The working mechanism of the proposed approach is not clear. While the empirical results show the effectiveness of this approach, more explanation of the mechanism is necessary. Specifically,

    **a)** Without additional constraint, why do the degradation network and restoration network correspond to degradation and restoration respectively? Is it possible that the degradation network attempts to enhance the HR input and the restoration network degrades back?

    **b)** Why does the learned degradation network reveal the real-world degradations? In theory, there could be unlimited possibilities. How does it work?

2. The claim in Ln.118 that `Almost all types of image degradation could ﬁnd a corresponding operation in a standard convolutional neural network.` is inaccurate. In practice, there are many degradations that cannot be represented by standard CNN operations. For example, the camera shot noise and read noise corrupt the input in the raw image domain, which cannot be represented by common operations. Similarly for multiplicative noise and JPEG compression. While CNN can be used to approximate the degradations, the above statement is over-claimed.

3. The ablation studies (Sec. 5) is confusing. The configuration in Table 2 is difficult to understand. Please explain the ablation studies clearly. Specifically,

    **a)** What is classical SR training?

    **b)** What is synthetic data augmentation?

    **c)** What is severe random style shift? (The difference to b) is the identity initialization, why does it related to style?)

**Questions:**

While the proposed approach achieves promising performance, the underlying mechanism and motivation are unclear. It is advised to dress the concerns in the weakness section. I would be happy to adjust the rating if the aforementioned concerns are resolved.

**Limitations:**

The authors addressed the limitations in the supplementary material.

---

> ### Author Rebuttal · Authors · 2023-08-10
>
> **Q1:** Why do the degradation network and restoration network correspond to degradation and restoration respectively? Is it possible that the degradation network attempts to enhance the HR input and the restoration network degrades back?
>
> **A1:** As demonstrated in Algorithm 1 of our supplementary material, the degradation network and restoration network are optimized alternatively. With a temporarily fixed restoration network, the degradation network continually seeks to increase the loss, intensifying image degradation. Conversely, with a temporarily fixed degradation network, the restoration network consistently endeavors to minimize the loss, enhancing image restoration quality. Therefore, we do not believe that the scenario where the degradation network enhances while the restoration network degrades would result in a stable convergence point. While this situation might appear feasible in theory, in practice, even a minor random disturbance could quickly propel the entire system out of that situation.
> ***
>
>
> **Q2:** Why does the learned degradation network reveal the real-world degradations? In theory, there could be unlimited possibilities. How does it work?
>
> **A2:** We do not require any perturbed degradation network to correspond exactly with a real-world degradation. This is neither necessary nor feasible for our method. We only need to guarantee that the entire feasible region of the degradation network could cover most cases of real-world degradation. In other words, we only need to guarantee that most instances of real-world degradation could be represented by a degradation network. If we can establish a solid lower performance bound for the entire feasible region of the degradation network through adversarial training, this lower bound also applies to most real-world degradation cases, thus achieving robust restoration.
> ***
>
> **Q3:** The claim that "Almost all types of image degradation could find a corresponding standard CNN operation" is inaccurate. There are many degradations that cannot be represented by standard CNN operations, such as noise in the raw image domain, multiplicative noise, and JPEG compression.
>
> **A3:** Thank you for pointing this out. We will revise the claim to make it more rigorous. We completely agree with your point that the degradations you mentioned cannot be accurately represented by standard CNN operations. In fact, we addressed similar issues in the Limitations section of our supplementary materials. The example of noise in the raw image domain, which you referred to, is similar to the halftoning degradation we discussed. Similarly, the scenario involving multiplicative noise closely resembles a low-light environment. Perhaps we could claim that "Most dominated image degradations in real-world SR tasks could find corresponding standard CNN operations"? If you have any suggestions, please don't hesitate to share with us.
>
> ***
> **Q4:** The ablation studies is confusing. The configuration in Table 2 is difficult to understand. Please explain the ablation studies clearly.
>
> **A4:** We apologize for any confusion that may have arisen. The ablation studies are intended to investigate the effects of the three major components of our method: adversarial perturbation, neural degradation, and identity initialization. In each ablation setting, we retain only specific components and assess the method's performance. The first column, labeled as "Configuration" in Table 2, is a name and an explanation for a particular ablation setting.
>
> If we do not use adversarial perturbation and neural degradation at all, our method would become a classical SR training method, which assumes that the image degradation model is an ideal bicubic downsampling. Most SR researches are with this setting, such as SRCNN, EDSR and RCAN.
>
> If we retain solely the adversarial perturbation without inducing neural degradation, it implies the utilization of an adversarial noise training method similar to "Generalized real-world super-resolution through adversarial robustness" (ICCVW2021). The perturbation under this setting is additive noise, following most adversarial training researches on image classification tasks.
>
> If we use neural degradation with identity initialization, but without adversarial perturbation, our method is then a SR training with synthetic data augmentation, working like "Designing a practical degradation model for deep blind image super-resolution" (ICCV2021). During model training, the neural degradation would be random sampling near the identity initialization rather than adversarial sampling, and the generated LR patches works like synthetic data augmentations.
>
> If we employ neural degradation with identity initialization, but exclude adversarial perturbation, our approach becomes analogous to SR training with synthetic data augmentation, akin to the method presented in "Designing a practical degradation model for deep blind image super-resolution" (ICCV 2021). Throughout the model training process, neural degradation involves random sampling near an identity mapping, rather than adversarial sampling. Consequently, the generated LR patches function as synthetic data augmentations.
>
> If we only remove the identity initialization from our method, the generated LR patches would no longer be visually similar to the HR patches. While the skeleton of the LR patches would remain unaffected, their color and texture would undergo a dramatic change, as the mapping would no longer be an identity mapping. This phenomenon is referred to as "severe random style shift" in our experiments.

---

> > ### Comment · Reviewer_Qwb9 · 2023-08-18
> >
> > Thank you for the authors' response. While the proposed method shows decent performance, it is non-trivial why this adversarial approach would converge to real-world degradations. Specifically, since there are unlimited ways to degrade an image, why would it correspond to real-world degradations? Given that this is a NeurIPS paper, it would be good to have more analysis and insights for this.

---

> > > ### Author Response · Authors · 2023-08-19
> > > **Reply to Reviewer Qwb9**
> > >
> > > Thank you for your response regarding the correspondence between the unlimited degradations that our method could generate and the real-world degradations. We addressed this question in our earlier rebuttal (Q2 and A2). It may help for us to clarify further.
> > >
> > > The degradations that our method can generate indeed have unlimited possibilities, forming an uncountably infinite set, denoted by $\mathbb{A}$. Similarly, the real-world degradations also form an uncountably infinite set, denoted by $\mathbb{B}$. It's evident that $\mathbb{A} \supset \mathbb{B}$.
> > >
> > >
> > > The adversarial training in our method solves a minmax problem on set $\mathbb{A}$, specifically, L = minmax (over set $\mathbb{A}$) loss. This process decreases the upper loss bound L, of the loss function within set $\mathbb{A}$. The upper loss bound for real-world degradations, denoted by L' = minmax (over set $\mathbb{B}$) loss, is lower than L, which means that the model always performs better on real-world degraded images during inference, than on highly adversarial degraded images during training.
> > >
> > > To give readers more concrete feelings for the validity of our adversarial degradation model, we will visualize LR images generated by our methods along with their corresponding kernels. As Reviewer 9QoZ and Reviewer vm11 suggested, we will visually compare them with real-world LR images, also quantitatively compare them in the Feature Frechet Distance.
> > >
> > > We hope that we have addressed your concerns, and we would greatly appreciate it if you could adjust your rating of our paper.

---

> > > > ### Comment · Reviewer_Qwb9 · 2023-08-20
> > > >
> > > > Thank you again for the response.
> > > >
> > > > - Why $\mathbb{B}$ $\subset$ $\mathbb{A}$? I believe there could be real-world degradations that cannot be produced by the proposed method, and there could also be degradations produced by the proposed method do not belong to real-world degradations. This claim should be carefully verified.
> > > >
> > > > - Using the above notations, even if $\mathbb{B}$ $\subset$ $\mathbb{A}$, my previous question is why the model converges to degradations $D\in \mathbb{A}\cap\mathbb{B}$, but not $D\in \mathbb{A}\setminus\mathbb{B}$?

---

> > > > > ### Author Response · Authors · 2023-08-20
> > > > > **Reply to Reviewer Qwb9**
> > > > >
> > > > > Thank you for the clarification of your questions.
> > > > >
> > > > > Q1: Why $\mathbb{B}$ $\subset$ $\mathbb{A}$? I believe there could be real-world degradations that cannot be produced by the proposed method.
> > > > >
> > > > > A1: This claim holds for all moderate degradations, as long as the image difference between the degraded image and the ground truth image is smaller than the representation limit of the noise injection layers. Taking JPEG compression as an example, the image artifacts can be represented by noise injection layers as additive noise, even if all other types of layers fail to contribute themselves. We acknowledge that severe degradations which drastically change the image cannot be produced by the proposed method, but they are rare among modern photographs in the real-world SR problem.
> > > > >
> > > > > Q2: Even if $\mathbb{B}$ $\subset$ $\mathbb{A}$, why the model converges to degradations $D\in \mathbb{A}\cap\mathbb{B}$, but not $D\in \mathbb{A}\setminus\mathbb{B}$?
> > > > >
> > > > > A2: The degradation model doesn't always converge to $D \in \mathbb{A} \cap \mathbb{B}$. Please note that we re-initialize the degradation model to the identity transformation before every optimization step of the restoration model (see Algorithm 1 on page 2 of the supplementary material). During each adversarial training phase for the degradation model, it might indeed converge to $D \in \mathbb{A} \setminus \mathbb{B}$, which means that our model produces a degradation that does not belong to real-world degradations. However, such situations indicate that the restoration model already performs well for $D \in \mathbb{A} \cap \mathbb{B}$, at least better than its performance for $D \in \mathbb{A} \setminus \mathbb{B}$. Therefore, the degradation model could converge to $D \in \mathbb{A} \setminus \mathbb{B}$; otherwise, it would converge to $D \in \mathbb{A} \cap \mathbb{B}$. So, even if that happens, we still achieve our goal, which is to make the restoration model perform well on real-world degradations.

---

> ### Author Response · Authors · 2023-08-21
> **Followup Response to Reviewer Qwb9**
>
> Dear Reviewer Qwb9,
>
> As the discussion period draws to a close within the next few hours, we are writing to inquire if your concerns have been successfully resolved. We genuinely hope our efforts might lead to a possible adjustment in the rating.
>
> Best regards,
>
> Authors

---

### Official Review · Reviewer_JnSR · 2023-07-06

**Soundness:** 3 good
**Presentation:** 3 good
**Contribution:** 3 good
**Rating:** 7
**Confidence:** 4

**Summary:**

This work proposed a novel blind SR method via adversarial neural degradation. Utilizing the proposed adversarial neural degradation model can generate various nonlinear degradations effects and no supervisions are required. This makes the proposed method can deal with various real SR datasets. Experiments also verify the effectiveness of the proposed method.

**Strengths:**

1.	Utilizing adversarial neural degradation model for degradation synthesis is novel.
2.	The performance is promising.


**Weaknesses:**

1.	In table 1, the authors are expected to provide the SR results with full supervisions by sota sr methods. The proposed method is not required to outperform these methods since they are full-supervised but the proposed method is zero-shot. This can help the readers know the gap between the proposed method and supervised methods.

**Questions:**

See the weaknesses.

**Limitations:**

The authors discussed the limitations.

---

> ### Author Rebuttal · Authors · 2023-08-10
>
> **Q1:** In table 1, the authors are expected to provide the SR results with full supervisions by sota sr methods. The proposed method is not required to outperform these methods since they are full-supervised but the proposed method is zero-shot. This can help the readers know the gap between the proposed method and supervised methods.
>
> **A1:** Thank you for your advice. We will incorporate it into the revised version of our paper. However, obtaining both degraded images and the corresponding latent images (ground truth) is expensive in reality. As a result, the number of images in real-world SR datasets is typically fewer than the number of images in the training set used in classical SR. We are curious if state-of-the-art SR methods trained with full supervision on such a small training set could serve as a reliable upper bound.

---

> > ### Comment · Reviewer_JnSR · 2023-08-20
> >
> > Yes, the number of training pairs in real-world SR dataset is fewer. Their testing performance can be treated as an overfitting to the current degradation. Therefore, the testing performance can serve as the upper bound for this specific degradation. The authors claimed that their model can cover most of the real-world degradations. I am curious about the performance of this model when compared with methods with full supervisions. The authors are expected to give this kind of comparison in rebuttal other than promising to give the results in the revision.

---

> > > ### Author Response · Authors · 2023-08-20
> > > **Reply to Reviewer JnSR**
> > >
> > > Thank you for your response regarding the comparison with methods with full supervision. We fully agree with your viewpoint that such an experiment can show the upper bounds of a specific degradation. RealSR [1] and DRealSR [2] are two real-world image SR dataset we used in our experiment. In Table 5 (page 12) of the DRealSR paper [2], the authors had already evaluated the performance of models with full supervision using real-world HR/LR pairs. As we adopted ESRGAN as our SR model and didn't alter its architecture, the data from their table can be directly applied here. We reorganized their data into the following table for simplicity. Additionally, we included the performance of DAN [3] in the table to provide you with a more concrete feeling of the gap between our method and the upper bound.
> > >
> > > | Method           | PSNR on RealSR [1] | PSNR on DRealSR [2] |
> > > | ---------------  | ----------------   | ----------------    |
> > > | DAN [3]          | 27.80              | 30.59               |
> > > | ANDNet (ours)    | 28.47              | 30.97               |
> > > | Full Supervision | 29.15              | 31.92               |
> > >
> > > [1] Toward real-world single image super-resolution: A new benchmark and a new model. ICCV 2019
> > >
> > > [2] Component divide-and-conquer for real-world image super-resolution. ECCV 2020
> > >
> > > [3] Unfolding the alternating optimization for blind super resolution. NeurIPS 2020

---

### Official Review · Reviewer_vm11 · 2023-07-10

**Soundness:** 3 good
**Presentation:** 3 good
**Contribution:** 3 good
**Rating:** 7
**Confidence:** 5

**Summary:**

This paper proposed a new image degradation system for blind image super-resolution tasks. The proposed method uses a neural network system to learn and capture the image degradation operations, combined with a image restoration network, the proposed method achieved satisfactory performance.

**Strengths:**

The proposed method introduced an image degradation system captured by a neural network system. The neural network is intuitive and has shown effectiveness on capture real-world degradation based on experimental results and analysis in the paper.

The presentation of the paper is good and the materials are well organized.

**Weaknesses:**

The paper is focusing on the degradation system and only adapted an existing SR model to be trained with the proposed system. It is not a weakness per see, but a new GAN system designed with the degradation system could have better performance.

**Questions:**

+ Can the author help visualize the degradation system's filter after the training, and compare it with real-world blur/noise/gamma kernel and effects?

**Limitations:**

The paper discussed some real-world use cases. The ablation study provides some insight into the system's different behavior.

---

> ### Author Rebuttal · Authors · 2023-08-10
>
> **Q1:** The paper is focusing on the degradation system and only adapted an existing SR model to be trained with the proposed system. It is not a weakness per see, but a new GAN system designed with the degradation system could have better performance.
>
> **A1:** Thank you for your advice. This work is inspired by two observations of image degradations. These two properties can be easily used in degradation model design, but for the time being, we cannot figure out how to use them in restoration model design. That is why we only adapted an existing SR model.
> ***
>
> **Q2:** Can the author help visualize the degradation system's filter after the training, and compare it with real-world blur/noise/gamma kernel and effects?
>
> **A2:** Thank you for your advice. We will include visualizations of the degradation system's filter in the revised version of our paper. However, for the current model, these visualizations can hardly contain any information. This is because there are several layers in the neural degradation, and each layer is only responsible for a very small portion of the degradation. We cannot simply combine them together due to the presence of non-linear activation layers between them. We intend to retrain the model with a much smaller degradation model, consisting of only two or three layers. This adjustment aims to make the visualizations more informative. If you have any suggestions, please do not hesitate to share them with us.

---

### Official Review · Reviewer_9QoZ · 2023-07-11

**Soundness:** 2 fair
**Presentation:** 2 fair
**Contribution:** 3 good
**Rating:** 6
**Confidence:** 5

**Summary:**

The paper presented a blind super-resolution algorithm, in which a neural network is used to represent image degradation, followed by the incorporation of adversarial learning mechanisms to study "hard cases". The technique to represent image degradation involves initializing a neural network, ensuring the initialization is an identity mapping, then inducing specific perturbations in the network's filtering and noise, resulting in irregular image degradation. Adversarial learning primarily involves monitoring difficult cases during training and subsequently increasing the penalty, establishing an adversarial approach.

**Strengths:**

The main contribution of the paper is the application of neural networks to express forms of degradation, which is relatively innovative. This approach is more intricate than previous higher-order degradation and random degradation. Adversarial training, to some extent, may be beneficial as it promotes generality, though its innovativeness is not notably outstanding.

**Weaknesses:**

Despite the novelty of using a neural network to represent degradation, there are several concerns.

First, a neural network represent degradation as a distribution, which actually has a range. An ICML paper [R1] pointed out that the range of this distribution could directly impact its performance and generalization abilities. The paper doesn't describe the distribution's range, nor is it clear how this range compares to that of other methods like RealESRGAN or BSRGAN.

[R1] Crafting Training Degradation Distribution for the Accuracy-Generalization Trade-off in Real-World Super-Resolution. ICML 2023.

Secondly, I have concerns about the experiment due to the possible limited range of degradation in the network. It may perform better in specific ranges, as RealESRGAN and BSRGAN may have a much wider range of degradation.

Relatedly, the paper doesn't discuss the network's generalization performance for degradation. If degradation is a range within a distribution, there must be a range, and neither enlarging nor narrowing it is inherently better. The paper does not offer any control for this range.

Lastly, the article's writing is problematic. Many descriptions are long and repetitive, like section 3.1 and 3.2, which could have been resolved with a simple diagram, but the verbose writing makes it difficult to communicate a straightforward idea.

Furthermore, the approach to representing degradation with a neural network seems intended to argue that this design of versatile degradation is more similar to real-life situations, particularly for low-resolution images that could be sampled from mobile ISPs. However, the paper lacks any descriptions or visualizations of low-resolution images and doesn't present experimental results to demonstrate the closeness of their design to actual cases. Hence, I have reservations about the nature of the low-resolution images they've created.

**Questions:**

See weakness.

**Limitations:**

Description is not clear

---

> ### Author Rebuttal · Authors · 2023-08-10
>
> Thank you for bringing the ICML 2023 paper to our attention, and we will discuss the contribution of the ICML paper in the revised version of our paper. As several of your questions are rooted in this paper, please let us first elaborate on the connection between the ICML paper and our research.
>
> The ICML paper indicates that when the degradation distribution during both training and testing perfectly matches, the SR model exhibits favorable generalization and achieves high performance simultaneously. The binning method is utilized to adjust the joint distribution of the three parameters of a widely-used image degradation model, helping the training degradation distribution to better match the testing distribution.
>
> We fully agree with the indication in the ICML paper regarding the importance of aligning training and testing degradation. However, we would like to emphasize that the ultimate objective should be the alignment of distributions across the set of degradation functions, whereas the ICML paper focuses solely on aligning the distributions of degradation parameters within a given model.
>
> This nuanced yet critical distinction can be elucidated through a simple analogy: Just as people cannot perfectly align a circle with a square by solely adjusting the circle's location and radius, they must also modify its shape. In this analogy, the location and radius correspond to the degradation parameters in the ICML paper, while modifying the shape corresponds to transitioning to a new family of functions, as demonstrated in our paper. In other word, if the blur-noise-JPEG model cannot generate LR image like the testing images, only using the binning method to adjust the distribution of parameters are not able to align the distributions. So our neural degradation model and the ICML paper both attempt to align degradation distributions. However, they focus on distinct and parallel areas.
>
> ***
> **Q1:** The paper doesn't describe the distribution's range, nor is it clear how this range compares to that of other methods like RealESRGAN or BSRGAN.
>
> **A1:** We have described the feasible region of our model in Section 4.3 of our paper and in Section A.2 of our supplementary material. However, because of the distinct nature of the function family employed in our paper compared to other methods, conducting a quantitative comparison of their ranges poses challenges. In the revised version of our paper, we will present a visual comparison of LR patches.
>
> ***
> **Q2:** The SR model may only perform better in limited range of degradation.
>
> **A2:** Your assertion is certainly true for any SR model, including ours, as the ICML paper indicates the importance of aligning training and testing degradation. However, we believe that the SR model which performs better within the real-world degradation range is more valuable than one that works better in another range. Quantitative comparison could demonstrate that our method generalizes better across all four real-world SR datasets. The range limitation is inherent to its nature rather than a drawback.
> ***
>
> **Q3:** The article's writing is problematic. Many descriptions are long and repetitive, like section 3.1 and 3.2.
>
> **A3:** Thank you for your advice. We will revise those two sections to make them more concise.
> ***
> **Q4:** The paper lacks visualizations of LR images.
>
> **A4:** Thank you for your advice. We will include visualizations of LR images in the revised version of our paper.
> ***
> **Q5:** Description of limitations is not clear
>
> **A5:** We did discuss the limitations of our method in the Limitations section of our supplementary material (page 3).

---

> > ### Comment · Reviewer_9QoZ · 2023-08-18
> > **Raise My Rating**
> >
> > After reading the author's rebuttal, I decided to Raise My Rating a bit. Some descriptions from the author make me think this paper is valuable. But I still think it can be done better experimentally.

---

> > > ### Author Response · Authors · 2023-08-19
> > > **Reply to Reviewer 9QoZ**
> > >
> > > Thank you for your response and your suggestion for strengthening the experimental part of our paper. We will include the following two new experiments to address your concerns in the revised version of our paper.
> > >
> > > First, to assess the performance of our method within a degradation range, we will compare it with other SOTA methods using synthetic HR/LR pairs. While the methods such as RealESRGAN or BSRGAN might work well in a large degradation range with a simple blur-noise-JPEG degradation model, we expect our model to outperform in cases of more complex but more realistic degradations, such as non-aligned double JPEG compression and degradations introduced by image enhancement processes.
> > >
> > > Second, we will visualize LR images generated by our methods along with their corresponding kernels. We will visually compare them with real-world LR images, and also quantitatively compare them in the Feature Frechet Distance, as outlined in the ICML paper.

---

### Decision · Program_Chairs · 2023-09-21

**Decision:**

Accept (poster)

**Comment:**

Before the discussion and rebuttal, reviewers highlighted the following strengths and weaknesses of Submission967:

*Strengths:*
- The use of adversarial neural degradation for degradation synthesis.
- Demonstrated effectiveness in capturing real-world degradation.
- Performance of the proposed method outperforms existing works on multiple datasets.
- Extensive experimental studies, including ablation, that showcase the method's effectiveness.

*Weaknesses:*
- The paper lacks clarity on the distribution range of degradation by the neural network, and its comparison to methods like RealESRGAN or BSRGAN.
- Questions regarding the experiment's possible limited range of degradation.
- Absence of discussions on the network's generalization performance for degradation.
- Issues with the paper's writing, including verbosity and lack of visualization.
- Unclear working mechanism of the proposed approach.
- Over-claims made in the paper, like the potential of CNNs to represent all types of image degradation.
- Confusing ablation studies.
- The idea of learning degradation for super-resolution isn't new, and the paper's contribution may not be significant.

Post-rebuttal, authors have addressed most concerns raised by reviewers. Reviewer 9QoZ acknowledges the paper's value, though suggests better experimental work. Reviewer Qwb9 seeks deeper insights into why the adversarial approach mirrors real-world degradations. Authors should tone down the claim regarding the capabilities of CNNs in representing all image degradation and show the degraded images in the revised version of the paper. Reviewer BEQG points out similarities to the ECCV2018 paper. The authors have addressed this concern by explaining the differences against the ECCV2018 paper. The AC concurred with the authors. Based on the overall comments, the decision is to accept the paper.